# LEARNING TO SIMULATE

**Nataniel Ruiz** [1,‡]**, Samuel Schulter** [2]**, Manmohan Chandraker** [2,3]
[1]Department of Computer Science, Boston University
[2]NEC Laboratories America
[3]UC San Diego
nruiz9@bu.edu, {samuel,manu}@nec-labs.com

## ABSTRACT

Simulation is a useful tool in situations where training data for machine learning models is costly to annotate or even hard to acquire. In this work, we propose a reinforcement learning-based method for automatically adjusting the parameters of any (non-differentiable) simulator, thereby controlling the distribution of synthesized data in order to maximize the accuracy of a model trained on that data. In contrast to prior art that hand-crafts these simulation parameters or adjusts only parts of the available parameters, our approach fully controls the simulator with the actual underlying goal of maximizing accuracy, rather than mimicking the real data distribution or randomly generating a large volume of data. We find that our approach (i) quickly converges to the optimal simulation parameters in controlled experiments and (ii) can indeed discover good sets of parameters for an image rendering simulator in actual computer vision applications.

## 1 INTRODUCTION

In order to train deep neural networks, significant effort has been directed towards collecting large-scale datasets for tasks such as machine translation (Luong et al., 2015), image recognition (Deng et al., 2009) or semantic segmentation (Geiger et al., 2013; Cordts et al., 2016). It is, thus, natural for recent works to explore simulation as a cheaper alternative to human annotation (Gaidon et al., 2016; Ros et al., 2016; Richter et al., 2016). Besides, simulation is sometimes the most viable way to acquire data for rare events such as traffic accidents. However, while simulation makes data collection and annotation easier, it is still an open question *what distribution* should be used to synthesize data. Consequently, prior approaches have used human knowledge to shape the generating distribution of the simulator (Sakaridis et al., 2018), or synthesized large-scale data with random parameters (Richter et al., 2016). In contrast, this paper proposes to *automatically determine simulation parameters* such that the performance of a model trained on synthesized data is maximized.

Traditional approaches seek simulation parameters that try to model a distribution that resembles real data as closely as possible, or generate enough volume to be sufficiently representative. By learning the best set of simulation parameters to train a model, we depart from the above in three crucial ways. First, the need for laborious human expertise to create a diverse training dataset is eliminated. Second, learning to simulate may allow generating a smaller training dataset that achieves similar or better performances than random or human-synthesized datasets (Richter et al., 2016), thereby saving training resources. Third, it allows questioning whether mimicking real data is indeed the best use of simulation, since a different distribution might be optimal for maximizing a test-time metric (for example, in the case of events with a heavy-tailed distribution).

More formally, a typical machine learning setup aims to learn a function $h_{\boldsymbol{\theta}}$ that is parameterized by $\boldsymbol{\theta}$ and maps from domain $\mathcal{X}$ to range $\mathcal{Y}$, given training samples $(\boldsymbol{x}, \boldsymbol{y}) \sim p(\mathbf{x}, \mathbf{y})$. Data $\boldsymbol{x}$ usually arises from a real world process (for instance, someone takes a picture with a camera) and labels $\boldsymbol{y}$ are often annotated by humans (someone describing the content of that picture). The distribution $p(\mathbf{x}, \mathbf{y})$ is assumed unknown and only an empirical sample $\boldsymbol{D} = \{(\boldsymbol{x}_i, \boldsymbol{y}_i)\}_{i=1}^{\mathrm{N}}$ is available.

The simulator attempts to model a distribution $q(\mathbf{x}, \mathbf{y}; \boldsymbol{\psi})$. In prior works, the aim is to adjust the form of $q$ and parameters $\boldsymbol{\psi}$ to mimic $p$ as closely as possible. In this work, we attempt to automatically

---

‡ This work was part of N. Ruiz's internship at NEC Labs America.

learn the parameters of the simulator $\psi$ such that the loss $\mathcal{L}$ of a machine learning model $h_{\boldsymbol{\theta}}$ is minimized over some validation data set $\boldsymbol{D}_{\text{val}}$. This objective can be formulated as the bi-level optimization problem

$$\boldsymbol{\psi}^* = \arg\min_{\boldsymbol{\psi}} \sum_{(\boldsymbol{x},\boldsymbol{y})\in\boldsymbol{D}_{\text{val}}} \mathcal{L}\left(y, h_{\boldsymbol{\theta}}(\boldsymbol{x}; \boldsymbol{\theta}^*(\boldsymbol{\psi}))\right) \tag{1a}$$

$$\text{s.t.} \quad \boldsymbol{\theta}^*(\boldsymbol{\psi}) = \arg\min_{\boldsymbol{\theta}} \sum_{(\boldsymbol{x},\boldsymbol{y})\in\boldsymbol{D}_{q(\mathbf{x},\mathbf{y}|\boldsymbol{\psi})}} \mathcal{L}\left(\boldsymbol{y}, h_{\boldsymbol{\theta}}(\boldsymbol{x},\boldsymbol{\theta})\right), \tag{1b}$$

where $h_{\boldsymbol{\theta}}$ is parameterized by model parameters $\boldsymbol{\theta}$, $\boldsymbol{D}_{q(\mathbf{x},\mathbf{y}|\boldsymbol{\psi})}$ describes a data set generated by the simulator and $\boldsymbol{\theta}(\boldsymbol{\psi})$ denotes the implicit dependence of the model parameters $\boldsymbol{\theta}$ on the model's training data and consequently, for synthetic data, the simulation parameters $\boldsymbol{\psi}$. In contrast to Fan et al. (2018), who propose a similar setup, we focus on the actual data generation process $q(\mathbf{x},\mathbf{y};\boldsymbol{\psi})$ and are not limited to selecting subsets of existing data. In our formulation, the upper-level problem (equation 1a) can be seen as a meta-learner that learns *how* to generate data (by adjusting $\boldsymbol{\psi}$) while the lower-level problem (equation 1b) is the main task model (MTM) that learns to solve the actual task at hand. In Section 2, we describe an approximate algorithm based on policy gradients (Williams, 1992) to optimize the objective 1. For our algorithm to interact with a black-box simulator, we also present an interface between our model's output $\boldsymbol{\psi}$ and the simulator input.

In various experiments on both toy data and real computer vision problems, Section 4 analyzes different variants of our approach and investigates interesting questions, such as: "Can we train a model $h_{\boldsymbol{\theta}}$ with less but targeted high-quality data?", or "Are simulation parameters that approximate real data the optimal choice for training models?". The experiments indicate that our approach is able to quickly identify good scene parameters $\boldsymbol{\psi}$ that compete and in some cases even outperform the actual validation set parameters for synthetic as well as real data, on computer vision problems such as object counting or semantic segmentation.

## 2 METHOD

### 2.1 PROBLEM STATEMENT

Given a simulator that samples data as $(\boldsymbol{x},\boldsymbol{y}) \sim q(\mathbf{x},\mathbf{y};\boldsymbol{\psi})$, our goal is to adjust $\boldsymbol{\psi}$ such that the MTM $h_{\boldsymbol{\theta}}$ trained on that simulated data minimizes the risk on real data $(\boldsymbol{x},\boldsymbol{y}) \sim p(\mathbf{x},\mathbf{y})$. Assume we are given a validation set from real data $\boldsymbol{D}_{\text{val}}$ and we can sample synthetic datasets $\boldsymbol{D}_{q(\mathbf{x},\mathbf{y}|\boldsymbol{\psi})} \sim q(\mathbf{x},\mathbf{y}|\boldsymbol{\psi})$. Then, we can can train $h_{\boldsymbol{\theta}}$ on $\boldsymbol{D}_{q(\mathbf{x},\mathbf{y}|\boldsymbol{\psi})}$ by minimizing equation 1b. Note the explicit dependence of the trained model parameters $\boldsymbol{\theta}^*$ on the underlying data generating parameters $\boldsymbol{\psi}$ in equation 1b. To find $\boldsymbol{\psi}^*$, we minimize the empirical risk over the held-out validation set $\boldsymbol{D}_{\text{val}}$, as defined in equation 1a. Our desired overall objective function can thus be formulated as the bi-level optimization problem in equation 1.

Attempting to solve it with a gradient-based approach poses multiple constraints on the lower-level problem 1b like smoothness, twice differentiability and an invertible Hessian (Bracken & McGill, 1973; Colson et al., 2007). For our case, even if we choose the model $h_{\boldsymbol{\theta}}$ to fulfill these constraints, the objective would still be non-differentiable as we (i) sample from a distribution that is parameterized by the optimization variable and (ii) the underlying data generation process (e.g., an image rendering engine) is assumed non-differentiable for the sake of generality of our approach. In order to cope with the above defined objective, we resort to policy gradients (Williams, 1992) to optimize $\boldsymbol{\psi}$.

### 2.2 LEARNING TO SIMULATE DATA WITH POLICY GRADIENTS

Our goal is to generate a synthetic dataset such that the main task model (MTM) $h_{\boldsymbol{\theta}}$, when trained on this dataset until convergence, achieves maximum accuracy on the test set. The test set is evidently not available during train time. Thus, the task of our algorithm is to maximize MTM's performance on the validation set by generating suitable data. Similar to reinforcement learning, we define a policy $\pi_{\omega}$ parameterized by $\omega$ that can sample parameters $\boldsymbol{\psi} \sim \pi_{\omega}$ for the simulator. The simulator can be seen as a generative model $G(\mathbf{x},\mathbf{y}|\boldsymbol{\psi})$ which generates a set of data samples $(\boldsymbol{x},\boldsymbol{y})$ conditioned on $\boldsymbol{\psi}$. We provide more details on the interface between the policy and the data generating function in the following section and give a concrete example for computer vision applications in Section 4. The

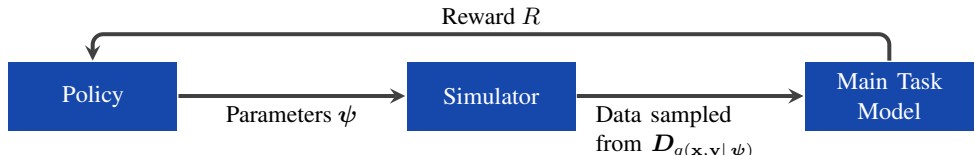

Figure 1: A high-level overview of our "learning to simulate" approach. A policy $\pi_\omega$ outputs parameters $\psi$ which are used by a simulator to generate a training dataset. The main task model (MTM) is then trained on this dataset and evaluated on a validation set. The obtained accuracy serves as reward signal $R$ for the policy on how good the synthesized dataset was. The policy thus learns *how to generate data to maximize the validation accuracy.*

policy receives a reward that we define based on the accuracy of the trained MTM on the validation set. Figure 1 provides a high-level overview.

Specifically, we want to maximize the objective

$$J(\omega) = \mathbb{E}_{\psi \sim \pi_\omega}[R] \tag{2}$$

with respect to $\omega$. The reward $R$ is computed as the negative loss $\mathcal{L}$ or some other accuracy metric on the validation set. Following the REINFORCE rule (Williams, 1992) we obtain gradients for updating $\omega$ as

$$\nabla_\omega J(\omega) = E_{\psi \sim \pi(\omega)} \big[ \nabla_\omega \log(\pi_\omega) R(\psi) \big] \ . \tag{3}$$

An unbiased, empirical estimate of the above quantity is

$$\mathcal{L}(\omega) = \frac{1}{K} \sum_{k=1}^{K} \nabla_\omega \log(\pi_\omega) \hat{A}_k \ , \tag{4}$$

where $\hat{A}_k = R(\psi_k) - b$ is the advantage estimate and $b$ is a baseline (Williams, 1992) that we choose to be an exponential moving average over previous rewards. In this empirical estimate, $K$ is the number of different datasets $\boldsymbol{D}_{q(\mathbf{x},\mathbf{y}|\psi_k)}$ sampled in one policy optimizing batch and $R(\psi_k)$ designates the reward obtained by the $k$-th MTM trained until convergence.

Given the basic update rule for the policy $\pi_\omega$, we can design different variants of our algorithm for learning to simulate data by introducing three control knobs. First, we define the number of training epochs $\xi$ of the MTM in each policy iteration as a variable. The intuition is that a reasonable reward signal may be obtained even if MTM is not trained until full convergence, thus reducing computation time significantly. Second, we define the size $M$ of the data set generated in each policy iteration. Third, we either choose to retain the MTM parameters $\boldsymbol{\theta}$ from the previous iteration and fine-tune on the newly created data or we estimate $\boldsymbol{\theta}$ from scratch (with a random initialization). This obviously is a trade-off because by retaining parameters the model has seen more training data in total but, at the same time, may be influenced by suboptimal data in early iterations. We explore the impact of these three knobs in our experiments and appendix. Algorithm 1 summarizes our approach.

---

**for** *iteration=1,2,...* **do**
    Use policy $\pi_\omega$ to generate $K$ model parameters $\psi_k$
    Generate $K$ datasets $\boldsymbol{D}_{q(\mathbf{x},\mathbf{y}|\psi_k)}$ of size $M$ each
    Train or fine-tune $K$ main task models (MTM) for $\xi$ epochs on data provided by $\mathcal{M}_k$
    Obtain rewards $R(\psi_k)$, i.e., the accuracy of the trained MTMs on the validation set
    Compute the advantage estimate $\hat{A}_k = R(\psi_k) - b$
    Update the policy parameters $\omega$ via equation 4
**end**

**Algorithm 1:** Our approach for "learning to simulate" based on policy gradients.

---

## 2.3 INTERFACING A SIMULATOR WITH THE POLICY

We defined a general black-box simulator as a distribution $G(\mathbf{x}, \mathbf{y} | \psi)$ over data samples $(\boldsymbol{x}, \boldsymbol{y})$ parameterized by $\psi$. In practice, a simulator is typically composed of a deterministic "rendering"

process $\mathcal{R}$ and a sampling step as $G(\mathbf{x}, \mathbf{y} \mid \boldsymbol{\psi}) = \mathcal{R}(S(\rho \mid \boldsymbol{\psi}), P(\phi \mid \boldsymbol{\psi}))$, where the actual data description $\rho$ (e.g., what objects are rendered in an image) is sampled from a distribution $S$ parametrized by the provided simulation parameters $\boldsymbol{\psi}$ and specific rendering settings $\phi$ (e.g., lighting conditions) are sampled from a distribution $P$ also parameterized by $\boldsymbol{\psi}$. To enable efficient sampling (via ancestral sampling) (Bishop, 2006), the data description distribution is often modeled as a Bayesian network (directed acyclic graph) where $\boldsymbol{\psi}$ defines the parameters of the distributions in each node, but more complex models are possible too.

The interface to the simulator is thus $\boldsymbol{\psi}$ which describes parameters of the internal probability distributions of the black-box simulator. Note that $\boldsymbol{\psi}$ can be modeled as an unconstrained continuous vector and still describe various probability distributions. For instance, a continuous Gaussian is modeled by its mean and variance. A K-dimensional discrete distribution is modeled with K real values. We assume the black-box normalizes the values to a proper distribution via a softmax.

With this convention all input parameters to the simulator are unconstrained continuous variables. We thus model our policy as the multivariate Gaussian $\pi_\omega(\rho, \phi) = \mathcal{N}(\omega, \sigma^2)$ with as many dimensions as the sum of the dimensions of parameters $\rho$ and $\phi$. For simplicity, we only optimize for the mean and set the variance to 0.05 in all cases, although the policy gradients defined above can handle both. Note that our policy can be extended to a more complicated form, e.g., by including the variance.

## 3 Discussion and related work

The proposed approach can be seen as a meta-learner that alters the data a machine learning model is trained on to achieve high accuracy on a validation set. This concept is similar to recent papers that learn policies for neural network architectures (Zoph & Le, 2016) and optimizers (Bello et al., 2017). In contrast to these works, we are focusing on the data generation parameters and actually create new, randomly sampled data in each iteration. While (Fan et al., 2018) proposes a subset selection approach for altering the training data, we are actually creating new data. This difference is important because we are not limited to a fixed probability distribution at data acquisition time. We can thus generate or oversample unusual situations that would otherwise not be part of the training data.

Similar to the above-mentioned papers, we also choose a variant of stochastic gradients (policy gradients (Williams, 1992)) to overcome the non-differentiable sampling and rendering and estimate the parameters of the policy $\pi_\omega$. While alternatives for black-box optimization exist, like evolutionary algorithms (Salimans et al., 2017) or sampling-based methods (Bishop, 2006), we favor policy gradients in this work for their sample efficiency and success in prior art.

Ganin et al. (2018) train a policy to generate a program that creates a copy of an input image. Similar to us, they use policy gradients to train the policy, although they use an adversarial loss to construct their reward. Again, Louppe & Cranmer (2017) seek to tune parameters of a simulator such that the marginal distribution of the synthetic data matches the distribution of the observed data. In contrast to both works, we learn parameters of a simulator that maximize performance of a main task model on a specific task. The learned distribution need not match the distribution of the observed data.

When relying on simulated data for training machine learning models, the issue of "domain gap" between real and synthetic data arises. Many recent works (Ganin et al., 2016; Chen et al., 2017; Tsai et al., 2018) focus on bridging this domain gap, particularly for computer vision tasks. Even if we are able to tune parameters perfectly, there exists a simulation-to-real domain gap which needs to be addressed. Thus, we believe the contributions of our work are orthogonal.

## 4 Experiments

The intent of our experimental evaluation is (i) to illustrate the concept of our approach in a controlled toy experiment (section 4.1), (ii) to analyze different properties of the proposed algorithm 1 on a high-level computer vision task (section 4.3) and (iii) to demonstrate our ideas on real data for semantic image segmentation (section 4.5).

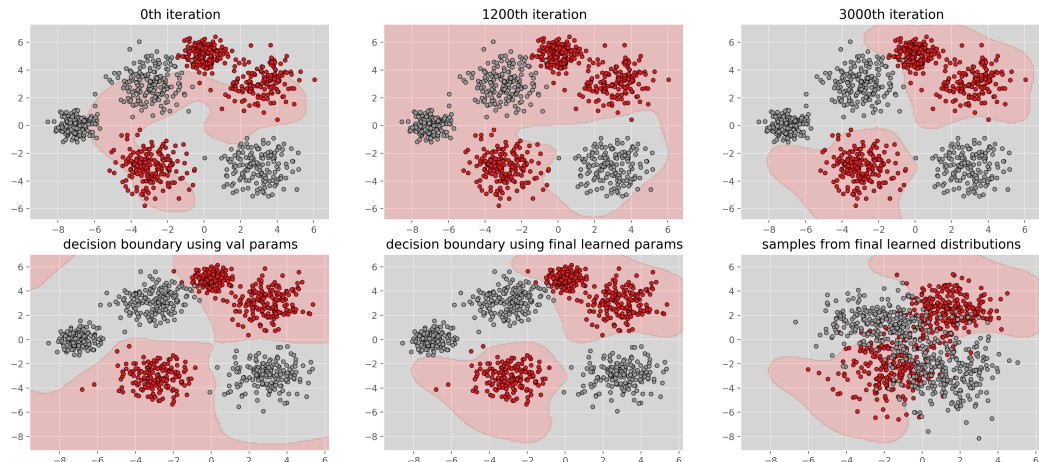

Figure 2: **Top row:** The decision boundaries (shaded areas) of a non-linear SVM trained on data generated by $q(\mathbf{x}, \mathbf{y} \,|\, \boldsymbol{\psi}_i)$ for three different iterations $i$ of our policy $\pi_\omega$. The data points overlaid are the test set. **Bottom row:** Decision boundary when trained on data sampled from $p(\mathbf{x}, \mathbf{y} \,|\, \boldsymbol{\psi}_{\text{real}})$ (left) and on the converged parameters $\boldsymbol{\psi}^*$ (middle); Data sampled from $q(\mathbf{x}, \mathbf{y} \,|\, \boldsymbol{\psi}^*)$ (right).

### 4.1 TOY EXPERIMENTS ON GAUSSIAN MIXTURES

To illustrate the concept of our proposed ideas we define a binary classification task on the 2-dimensional Euclidean space, where data distribution $p(\mathbf{x}, \mathbf{y} \,|\, \boldsymbol{\psi}_{\text{real}})$ of the two classes is represented by Gaussian mixture models (GMM) with 3 components, respectively. We generate validation and test sets from $p(\mathbf{x}, \mathbf{y} \,|\, \boldsymbol{\psi}_{\text{real}})$. Another GMM distribution $q(\mathbf{x}, \mathbf{y} \,|\, \boldsymbol{\psi})$ reflects the simulator that generates training data for the main task model (MTM) $h_{\boldsymbol{\theta}}$, which is a non-linear SVM with RBF-kernels in this case. To demonstrate the practical scenario where a simulator is only an approximation to the real data, we fix the number of components per GMM in $\boldsymbol{\psi}$ to be only 2 and let the policy $\pi_\omega$ only adjust mean and variances of the GMMs. Again, the policy adjusts $\boldsymbol{\psi}$ such that the accuracy (i.e., reward $R$) of the SVM is maximized on the validation set.

The top row of figure 2 illustrates how the policy gradually adjusts the data generating distribution $q(\mathbf{x}, \mathbf{y} \,|\, \boldsymbol{\psi})$ such that reward $R$ is increased. The learned decision boundaries in the last iteration (right) well separate the test data. The bottom row of figure 2 shows the SVM decision boundary when trained with data sampled from $p(\mathbf{x}, \mathbf{y} \,|\, \boldsymbol{\psi}_{\text{real}})$ (left) and with the converged parameters $\boldsymbol{\psi}^*$ from the policy (middle). The third figure in the bottom row of figure 2 shows samples from $q(\mathbf{x}, \mathbf{y} \,|\, \boldsymbol{\psi}^*)$. The sampled data from the simulator is clearly different than the test data, which is obvious given that the simulator's GMM has less components per class. However, it is important to note that the decision boundaries are still learned well for the task at hand.

### 4.2 GENERATING DATA WITH A PARAMETERIZED TRAFFIC SIMULATOR

For the following experiments we use computer vision applications and thus require a generative scene model and an image rendering engine. We focus on traffic scenes as simulators/games for this scenario are publicly available (CARLA (Dosovitskiy et al., 2017) with Unreal engine (Epic-Games, 2018) as backend). However, we need to note that substantial extensions were required to actually generate different scenes according to a scene model rather than just different viewpoints of a static map. Many alternative simulators like (Mueller et al., 2017; Richter et al., 2016; Shah et al., 2018) are similar where an agent can navigate a few pre-defined maps, but the scene itself is not parameterized and cannot be changed on the fly.

To actually synthesize novel scenes, we first need a model $S(\rho \,|\, \boldsymbol{\psi})$ that allows to sample instances of scenes $\rho$ given parameters $\boldsymbol{\psi}$ of the probability distributions of the scene model. Recall that $\boldsymbol{\psi}$ is produced by our learned policy $\pi_\omega$.

Our traffic scene model $S(\rho \,|\, \boldsymbol{\psi})$ handles different types of intersections, various car models driving on the road, road layouts and buildings on the side. Additionally our rendering model $P(\phi \,|\, \boldsymbol{\psi})$ handles

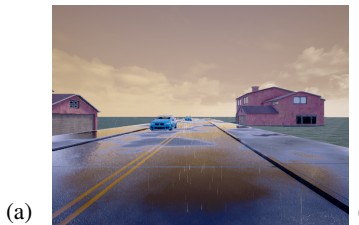
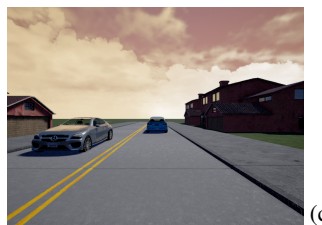
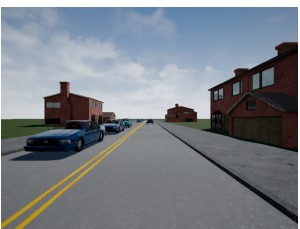

(a)                    (b)                  (c)

Figure 3: Example of rendered traffic scene with CARLA (Dosovitskiy et al., 2017) and the Unreal engine (Epic-Games, 2018).

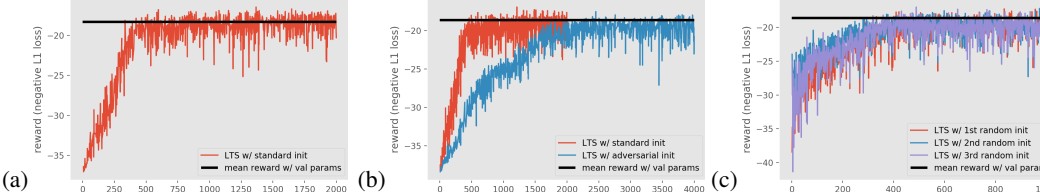

Figure 4: **(a)** shows the reward on the validation set evolving over policy iterations. **(b)** shows the reward on the unseen test set. We observe that even when using the "adversarial" initialization of parameters, our approach converges to the same reward $R$, but at a slower rate. **(c)** shows the reward on the unseen test for different random parameter initializations.

weather conditions. Please see the appendix for more details. In our experiments, the model is free to adjust some of the variables, e.g., the probability of cars being on the road, weather conditions, etc. Given these two distributions, we can sample a new scene and render it as $\mathcal{R}(S(\rho \,|\, \boldsymbol{\psi}), P(\phi \,|\, \boldsymbol{\psi}))$. Figure 3 shows examples of rendered scenes.

### 4.3 LEARNING TO SIMULATE ON A HIGH-LEVEL VISION TASK

As a first high-level computer vision task we choose counting cars in rendered images, where the goal is to train a convolutional neural network $h_{\boldsymbol{\theta}}$ to count all instances individually for five types of cars in an image. The evaluation metric and (also the loss) is the $\ell_1$ distance between predicted and ground truth count, averaged over the different car types. The reward $R$ is the negative $\ell_1$ loss. For this experiment, we generate validation and test sets with a fixed and pre-defined distribution $\boldsymbol{\psi}_{\text{real}}$.

**Initialization:** We first evaluate our proposed policy (dubbed "LTS" in the figures) for two different initializations, a "standard" random one and an initialization that is deliberately picked to be suboptimal (dubbed "adversarial"). We also compare with a model trained on a data set sampled with $\boldsymbol{\psi}_{\text{real}}$, i.e., the test set parameters. Figure 4 explains our results. We can see in figure 4a that our policy $\pi_{\omega}$ ("LTS") quickly reaches high reward $R$ on the validation set, equal to the reward we get when training models with $\boldsymbol{\psi}_{\text{real}}$. Figure 4b shows that high rewards are also obtained on the unseen test set for both initializations, albeit convergence is slower for the adversarial initialization. The reward after convergence is comparable to the model trained on $\boldsymbol{\psi}_{\text{real}}$. Since our policy $\pi_{\omega}$ is inherently stochastic, we show in figure 4c convergence for different random initializations and observe a very stable behavior.

**Accumulating data:** Next, we explore the difference between training the MTM $h_{\boldsymbol{\theta}}$ from scratch in each policy iteration or retaining its parameters and fine-tune, see algorithm 1. We call the second option the "accumulated main task model (AMTM)" because it is not re-initialized and accumulates information over policy iterations. The intention of this experiment is to analyze the situation where the simulator is used for generating large quantities of data, like in (Richter et al., 2016). First, by comparing figures 4b and 5a, we observe that the reward $R$ gets significantly higher than when training MTM from scratch in each policy iteration. Note that we still use the MTM reward as our training signal, we only observe the AMTM reward for evaluation purposes.

For the case of accumulating the MTM parameters, we further compare with two baselines. First, replicating a hand-crafted choice of simulation parameters, we assume no domain expertise and

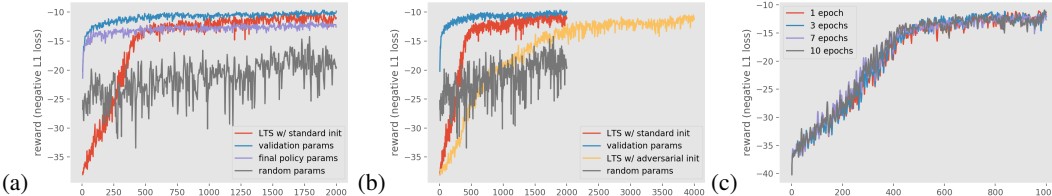

Figure 5: **(a)** Reward $R$ of the accumulated main task model on the car-counting task using different training schemes. We observe that training an accumulated main task network using a learning policy at each step is superior to training it with a dataset generated either using the final parameters of the policy or random parameters. **(b)** Learning-to-simulate converges even with an "adversarial" initialization of parameters, albeit in more epochs. **(c)** Reward of the accumulated main task model using different number of training epochs $\xi$ for $h_{\boldsymbol{\theta}}$.

randomly sample simulator parameters (within a sensible range) in each iteration ("random policy params"). Second, we take the parameters given by our learned policy after convergence ("final policy params"). For reference, we train another AMTM with the ground truth validation set parameters ("validation params") as our upper-bound. All baselines are accumulated main task models, but with fixed parameters for sampling data, i.e., resembling the case of generating large datasets. We can see from figure 5a that our approach gets very close to the ground truth validation set parameters and significantly outperforms the random parameter baseline. Interestingly, "LTS" even outperforms the "final policy params" baseline, which we attribute to increased variety of the data distribution. Again, "LTS" converges to a high reward $R$ even with an adversarial initialization, see figure 5b

**Number of epochs:** Similar to the previous experiment, we now analyze the impact of the number of epochs $\xi$ used to train the main task model $h_{\boldsymbol{\theta}}$ in the inner loop of learning to simulate. Figure 5c shows the reward of the accumulated MTM for four different values of $\xi$ (1, 3, 7, 10). Our conclusion, for the car-counting task, is that learning to simulate is robust to lower training epochs, which means that even if the MTM has not fully converged yet the reward signal is good enough to provide guidance for our system leading to a potential wall-time speed up of the overall algorithm. All four cases converge, including the one where we train the MTM for only one epoch. Note that this is dependent on the task at hand, and a more complicated task might necessitate convergence of the main task model to provide discriminative rewards.

## 4.4 SEMANTIC SEGMENTATION ON SIMULATED DATA

For the next set of experiments we use semantic segmentation as our test bed, which aims at predicting a semantic category for each pixel in a given RGB image (Chen et al., 2018). Our modified CARLA simulator (Dosovitskiy et al., 2017) provides ground truth semantic segmentation maps for generated traffic scenes, including categories like road, sidewalk or cars. For the sake of these experiments, we focus on the segmentation accuracy of cars, measured as intersection-over-union (IoU), and allow our policy $\pi_{\omega}$ to adjust scene and rendering parameters to maximize reward $R$ (i.e., car IoU). This includes the probability of generating different types of cars, length of road and weather type. The main task model $h_{\boldsymbol{\theta}}$ is a CNN that takes a rendered RGB image as input and predicts a per-pixel classification output.

We first generate validation set parameters $\psi_{\text{val}}$ that reflect traffic scenes moderately crowded with cars, unbalanced car types, random intersections and buildings on the side. As a reference point for our proposed algorithm, we sample a few data sets with the validation set parameters $\psi_{\text{val}}$, train MTMs and report the maximum reward (IoU of cars) achieved. We compare this with our learned policy $\pi_{\omega}$ and can observe in figure 6a that it actually outperforms the validation set parameters. This is an interesting observation because it shows that the validation set parameters $\psi_{\text{val}}$ may not always be the optimal choice for training a segmentation model.

## 4.5 SEMANTIC SEGMENTATION ON REAL DATA

We demonstrate the practical impact of our learning-to-simulate approach on semantic segmentation on KITTI Geiger et al. (2013) by training a main task model (MTM) $h_{\boldsymbol{\theta}}$ with a *reward signal coming from real data*. Using simulated data for semantic segmentation was recently investigated from a

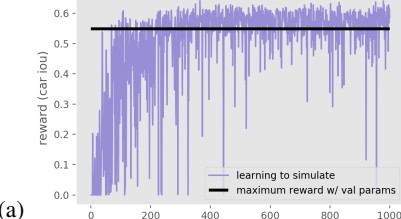 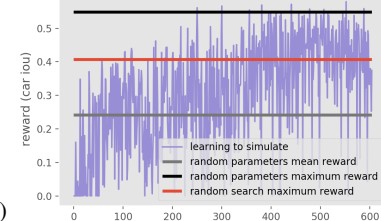

(a)                                                              (b)

Figure 6: **(a)** Reward curves of our approach compared to a model trained on data generated with the actual validation set parameters on the synthetic semantic segmentation task. **(b)** Reward curves on the real validation set of KITTI for semantic segmentation. We plot the learning-to-simulate, the maximum reward achieved using random search and the maximum and mean using random parameters. All methods use 600 iterations.

| Training data | random params | random search | LTS | KITTI train set |
|---|---|---|---|---|
| **Car IoU** | 0.480 | 0.407 | **0.579** | **0.778** |

Table 1: Segmentation Car IoU on the unseen KITTI test set for a ResNet-50 segmentation network trained using synthetic data generated by random parameters or learned parameters using random search or learning to simulate (LTS) for 600 epochs of each. We test the epoch with highest validation reward on the KITTI test set. We also report the maximum car IoU obtained by training on 982 annotated real KITTI training images.

domain adaptation perspective Tsai et al. (2018); Richter et al. (2016), where an abundant set of simulated data is leveraged to train models applicable on real data. Here, we investigate targeted generation of simulated data and its impact on real data. Since the semantic label space of KITTI and our CARLA simulator are not identical, we again focus on segmentation of cars by measuring IoU for that category. For our main task model $h_{\boldsymbol{\theta}}$ we use a CNN that takes a rendered RGB image as input and predicts a per-pixel classification output with a ResNet-50 backbone.

As our baseline we train the main task model separately 600 times, with data generated by the simulator using different sets of random parameters for each one. We monitor the validation Car IoU metric for each of these networks and pick the one with highest validation reward. We then test it on the unseen KITTI test set and report the Car IoU in table 1. For illustration purposes we show the reward curve of our approach on the validation set as well as the maximum for random search and the maximum and mean for random parameters in Figure 6b.

However, it is important to mention that parameters which are actually good for training an MTM $h_{\boldsymbol{\theta}}$ are unknown, making our automatic approach attractive in such situations. The results on the unseen real KITTI *test set* in table 1 confirm the superior results of learning-to-simulate. We train using synthetic data generated by random or learned parameters for 600 epochs of each. We pick the epoch with highest validation reward and test it on the KITTI test set. For reference, we also report the maximum car IoU obtained by our network by training on 982 annotated real KITTI training images.

Additionally, we verify empirically that parameter optimization using policy gradients (learning to simulate) outperforms random search for this problem. Results are reported in table 1.

## 5 CONCLUSION

Learning to simulate can be seen as a meta-learning algorithm that adjusts parameters of a simulator to generate synthetic data such that a machine learning model trained on this data achieves high accuracies on validation and test sets, respectively. Given the need for large-scale data sets to feed deep learning models and the often high cost of annotation and acquisition, we believe our approach is a sensible avenue for practical applications to leverage synthetic data. Our experiments illustrate the concept and demonstrate the capability of learning to simulate on both synthetic and real data. For future work, we plan to expand the label space in our segmentation experiments, apply the algorithm to other tasks like object detection and to explore a dynamic memory of previously generated data for improving our learning to simulate procedure.

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

| Gaussians per Class | Accuracy |
|---|---|
| 1 | 0.670 |
| 2 | **0.996** |
| val params | 0.995 |

Table 2: Accuracies on toy test set with learned $Q$ distributions with differing number of gaussians.

## A  TRAFFIC SCENE MODEL

Our model comprises the following elements:

- A straight road of variable length.
- Either an L, T or X intersection at the end of the road.
- Cars of 5 different types which are spawned randomly on the straight road.
- Houses of a unique type which are spawned randomly on the sides of the road.
- Four different types of weather.

All of these elements are tied to parameters: $\rho_k$ can be decomposed into parameters which regulate each of these objects. The scene is generated "block" by "block". A block consists of a unitary length of road with sidewalks. Buildings can be generated on both sides of the road and cars can be generated on the road. $\rho_{k,car}$ designates the probability of car presence in any road block. Cars are sampled block by block from a Bernouilli distribution $X \sim \text{Bern}(\rho_{k,car})$. To determine which type of car is spawned (from our selection of 5 cars) we sample from a Categorical distribution which is determined by 5 parameters $\rho_{k,car_i}$ where $i$ is an integer representing the identity of the car and $i \in [1,5]$. $\rho_{k,house}$ designates the probability of house presence in any road block. Houses are sampled block by block from a Bernouilli distribution $X \sim \text{Bern}(\rho_{k,house})$.

Length to intersection is sampled from a Categorical distribution determined by 10 parameters $\rho_{k,length_i}$ with $i \in [8,18]$ where $i$ denotes the length from the camera to the intersection in "block" units. Weather is sampled randomly from a Categorical distribution determined by 4 parameters $\phi_{k,weather_i}$ where $i$ is an integer representing the identity of the weather and $i \in [1,4]$. L, T and X intersections are sampled randomly with equal probability.

## B  QUANTITATIVE RESULTS ON TOY EXPERIMENT

In table 2, we present classification accuracy for the toy problem in Section 4.1 with $Q$ distributions using different number of gaussians. We can observe that by using learning to simulate we obtain better classification results than using a dataset generated using the test set parameters (mean and variance of gaussians in $P$ distribution).

## C  LEARNING OF PARAMETERS IN CAR COUNTING

In this section we visualize learning of parameters in the car counting problem described in Section 4.3. In particular we show how the parameters of weather type and car type evolve in time in Figure 7.

## D  CAR COUNTING DATASET SIZE

We explore the parameter $M$ of our algorithm that controls the dataset size generated in each policy iteration. For example, when $M = 100$, we generate at each policy step a dataset of 100 images using the parameters from the policy which are then used to train our main task model $h_{\boldsymbol{\theta}}$. We evaluate policies with sizes 20, 50, 100 and 200. In figure 8 we show a comparative graph of final errors on the validation and test sets for different values of $M$. For a fair comparison, we generate 40,000 images with our final learned set of parameters and train $h_{\boldsymbol{\theta}}$ for 5 epochs and evaluate on the test set. We observed that for this task a dataset size of just 20 suffices for our model to converge to good scene

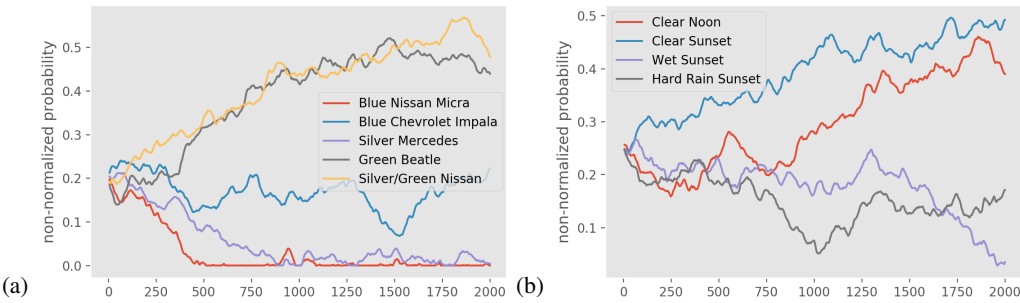

Figure 7: **(a)** Shows the evolution of the non-normalized probabilities (logits) of spawning different car types in time while the parameters are learned using our method in the car counting task. **(b)** Shows the evolution of the non-normalized probabilities (logits) of rendering different weather types in time.

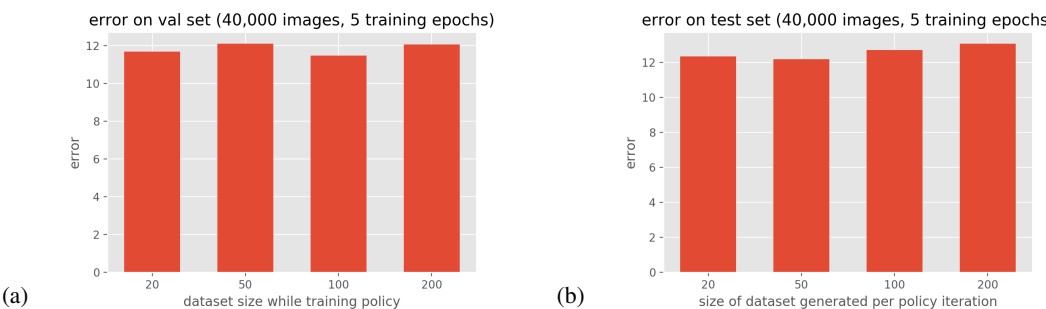

Figure 8: Test and validation error from main task networks trained on 40,000 images during 5 epochs using final learned parameters for different sizes of datasets generated per policy iteration.

parameters $\psi$, which is highly beneficial for the wall-time convergence speed. Having less data per policy iteration means faster training of the MTM $h_{\boldsymbol{\theta}}$.

## E   REPRODUCIBILITY

Since our method is stochastic in nature we verify that "learning to simulate" converges in the car counting task using different random seeds. We observer in figure 9a that the reward converges to the same value with three different random seeds. Additionally, in figure 9b, we observe that the accumulated main task network test reward also converges with different random seeds.

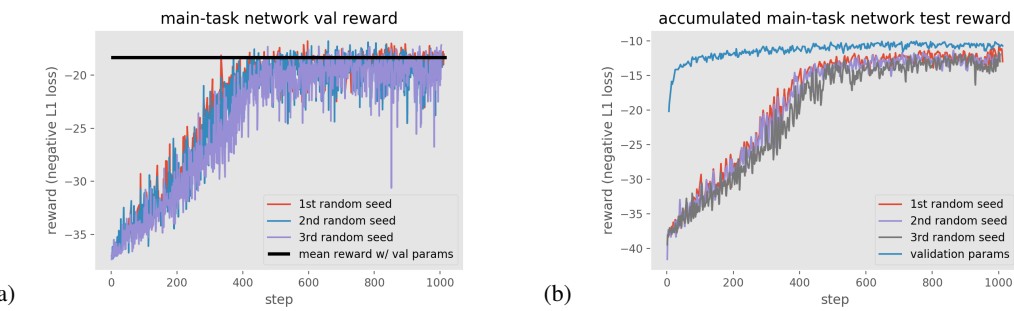

Figure 9: Main task network reward and accumulated MTN reward converge using different random seeds.

