# OpenReview forum: "Learning To Simulate"
_ICLR.cc/2019/Conference_

### Official Review · AnonReviewer3 · 2018-10-27
**stimulating idea; potentially flawed method and incomplete evaluation**

**Rating:** 7
**Confidence:** 4

**Review:**

The paper explores an interesting idea: automatically tuning the parameters of a simulation engine to maximize the performance of a model that is trained using this simulation engine. In the most interesting scenario, the model is trained using such optimized simulation and then tested on real data; this scenario is explored in Section 4.5.

The basic idea of optimizing simulation parameters for transfer performance on real data is very good. I believe that this idea will be further explored and advanced in future work. The present submission is either the first or one of the first papers to explicitly explore this idea, and deserves some credit and goodwill for this reason. This is the primary reason my rating is "marginally above acceptance threshold" and not lower.

The paper suffers from some issues in the technical formulation and experimental evaluation. The issues are reasonably serious. First, it is not clear at all that RL is the right approach to this optimization problem. There is no multi-step decision making, there are no temporal dynamics, there is no long-term credit assignment. The optimization problem is one-shot: you pick a set of parameters and get a score. Once. That's it. It's a standard black-box optimization setting with no temporal aspect. My interpretation is that RL is used here because it's fashionable, not because it's appropriate.

The evaluation is very incomplete and unsatisfactory. Let's focus on Table 1, which I view as the main result since it involves real data. First, is the optimization performed using the KITTI validation set? Without any involvement of the test set during the optimization? I hope so, but would like the authors to explicitly confirm.

Second, the only baseline, "random params", is unsatisfactory. I take this baseline to be the average performance of randomized simulation. But this is much too weak. Since the authors have access to the validation set during the optimization, they can simply test which of the random parameter sets performs best on the validation set and use that. This would correspond to the *best* set of parameters sampled during training. It's a valid baseline, there is no reason not to use it. It needs to be added to Table 1.

Also, 10 sets of random params seems quite low. How many sets of parameters does the RL solver sample during training? That would be the appropriate number of sets of params to test for the baseline. (And remember to take the *best* of these for the baseline.)

The last few points really boil down to setting up an honest random search baseline. I consider this to be mandatory and would like to ask that the authors do this for the rebuttal. There are also other derivative-free optimization techniques, and a more thorough evaluation would include some of these as well.

My current hypothesis is that an honest random search baseline will do as well as or better than the method presented in the submission. Then the submission boils down to "let's automatically tune simulation parameters; we can do this using random search". It's still a stimulating idea. Is it sufficient for an ICLR paper? Not sure. Something for the reviewers and the ACs to discuss as a group.

---

> ### Author Response · Authors · 2018-11-25
> **Added more extensive experiments and proposed baselines**
>
> We thank the reviewer for noting the novelty. We hope the following clarifications and new experiments ease their concerns.
>
> Use of policy gradients:
> We agree that our problem is a black box optimization without temporality or discounted reward. In Section 3, second paragraph, we do discuss alternatives such as evolutionary algorithms or sampling methods. Our use of policy gradients is not due to it being fashionable. Rather, we use them to estimate gradients for a non-differentiable function with the following advantages:
> (a) Simplicity: The method is simple, easy to implement and easy to formalize (see Algorithm 1). This makes it easily reproducible. There are very few hyperparameters to tune (baseline, learning rate of policy)
> (b) Flexibility: The policy that is defined can be arbitrarily flexible. We use a Gaussian policy but the work can be extended to discrete policies or those using neural networks.
> (c) Sample efficiency: We observe in all experiments that parameters converge after less than 500 iterations. For some experiments we observe convergence in less than 200 iterations. This is due to the direct relationship between our reward and the value we want to optimize (validation accuracy). In our case, those two are the same.
> (d) Interpretability: We show curves of weather probabilities and car type probabilities in a new figure in the appendix. We can visualize how the probabilities are learned through iterations.
>
> We believe that (a) and (b) are the most distinct advantages of policy gradients. (c) and (d) are advantages to a lesser extent and can be present in other derivative-free optimization methods. The generality afforded by the method is important since our work is designed to be applied to different applications where simulation is possible.
>
> We note that policy gradients are also used in other works that have a similar one-shot scenario as ours, such as “Neural Architecture Search”, “Neural Optimizer Search”, as well as both the works cited by Reviewer 1.
>
> Evaluation (test set):
> We emphasize the test set is not used in any experiment for parameter tuning. It is unseen for every problem and only used once for final evaluation. Section 2.2 and Figure 4 state this. We have modified the paper to highlight this further.
>
> Evaluation (comparison with “best” random sample):
> We already present a strong demonstration on the car counting task in Figure 5, where learning to simulate outperforms the “best” random parameters. We initialize two networks and train them using datasets generated by learning to simulate policy (red curve) and random policy (grey curve). We show that the random policy is vastly outperformed by the learned policy.
>
> Additionally we present a more extensive and fair real data segmentation experiment. We use a more powerful ResNet-50 backbone (ResNet-18 was used in the original submission for faster experimentation) and let our policy learn for 600 iterations and sample random parameters for 600 iterations. Our best policy iteration achieves 0.579 IoU, which is 20% better than the best dataset generated with random parameters (0.480 IoU). Thus, we show a clear improvement over this baseline. Our intuition is that the higher the dimensionality of the parameter space and smaller the areas of high reward, the more likely random parameters will have difficulty achieving high reward.
>
> Random search baseline as opposed to policy gradients:
> Thank you for this suggestion. We believe random search is a valid baseline, but not as sample efficient or successful as policy gradients in some scenarios. To verify this, we use a hypersphere radius of 0.1 for random search, extensively tuned using several runs of the method, for both the car counting and KITTI segmentation experiments. For car counting, which presents a less noisy reward signal, random search performs about the same as our method achieving an L1 error of 16.53 reward compared to 16.94 for our proposed method. However, for KITTI car segmentation, it achieves an IoU of 40.7% (using the same number of iterations, namely 600), yet policy gradients achieve higher IoU of 57.9%. In this scenario policy gradients demonstrates an increase in performance of 42%. This has been added to the paper.
>
> KITTI segmentation evaluation:
> Please see response to Reviewer 2, where we demonstrate 20% improvements over the best random parameters, 42% improvements over a well-tuned random search baseline, as well as obtaining IoU with 100 synthetic images that is reasonable compared to 982 real images for training.

---

> ### Comment · AnonReviewer3 · 2018-12-09
> **paper has improved; authors responded well to reviews**
>
> I upgraded my score from 6 to 7.
>
> The revision and the responses provided by the authors address some of my concerns.
>
> I still have doubts about the use of RL here. (I don't think it's needed.) And I wish the authors have gone further in the aspects of the simulation they optimize as well as the downstream tasks they tackle. Overall, on the methodological and the experimental fronts, I consider the paper to be rather weak. However, this is counterbalanced by the idea itself, which I find timely and stimulating. This paper may spur others to study this direction, bring more appropriate methods to bear on this problem, and attack more complex and realistic downstream tasks.
>
> As a kind of "lightning rod" that attracts attention and stimulates follow-up work, this paper can be a useful addition to the literature.
>
> I also appreciate that the authors have thoroughly addressed the reviewers' concerns and have added more substantial experimental results to the revision.
>
> Overall, the benefits of publishing the work probably outweigh the drawbacks.

---

### Official Review · AnonReviewer1 · 2018-11-02
**Sound method, but lacking a proper evaluation and comparison**

**Rating:** 6
**Confidence:** 5

**Review:**

This work makes use of policy gradients for fitting the parameters of a simulator in order to generate training data that results in maximum performance on real test data (e.g., for classification). The difficulty of the task rises from the non-differentiability of the simulator.

# Quality

The method is sound, well-motivated, and presented with a set of reasonable experiments. However, and this is a critical weakness of the paper, no attempt is made to compare the proposed method with respect to any related work, beyond a short discussion in Section 3. The experiments do include some baselines, but they are all very weak.

# Clarity

The paper is well-written and easy to follow. The method is illustrated with various experiments that either study some properties of the algorithm or show some good performance on real data.

# Originality

The related work is missing important previous papers that have proposed very similar/identical algorithms for fitting simulator parameters in order to best reproduce observed data. For example,
- https://arxiv.org/abs/1804.01118
- https://arxiv.org/abs/1707.07113
which both make use of policy gradients for fitting an adversary between fake and real data (which is then used a reward signal for updating the simulator parameters).

# Significance

The significance of the paper is moderate given some similar previous works. However, the significance of the method itself (regardless of previous papers) is important.

---

> ### Author Response · Authors · 2018-11-25
> **Added discussion on related work and points on sufficiency of comparison**
>
> We thank the reviewer for their comments. We provide additional evaluations and discussion of related works to address their concerns.
>
> Sufficiency of evaluation:
> We agree that a practical implementation would use a more extensive simulator, but we believe our choices sufficiently illustrate the idea, while keeping the effort reasonable for an ICLR paper. Please refer to the first three points in the response to Reviewer 2.
>
> Comparative references:
> Thank you for pointing out these papers. We briefly highlight below that our contributions are quite different from both of those works. We have added this discussion to the related work section.
>
> “Synthesizing Programs for Images using Reinforced Adversarial Learning” from ICML 2018 trains a policy to generate a program that creates a copy of the input image.
> (a) Simulators used in the paper are a brushstroke simulator and an object placer.
> (b) Some similarities are that they use reinforcement learning to update the parameters of their policy and generate synthetic data using a non-differentiable simulator.
> (c) But they generate plausible synthetic data identical to the input, or sample from the latent space to create a program that simulates an unconditioned sample. In contrast, we learn parameters of a simulator that maximize performance of a main task model.
> (d) Importantly, we do not wish to reproduce observed data. Indeed, the reward function can even be chosen to amplify some rare cases. For example, if an object category is rare in road scenes, but important to segment for collision avoidance, our reward can be used to reflect this.
> (e) Another difference is that they use an adversarial loss to create their reward signal, while we use the validation accuracy of the main task model.
>
> “Adversarial Variational Optimization of Non-Differentiable Simulators” is, to the best of our knowledge, an unpublished work. It seeks to "match the marginal distribution of the synthetic data to the empirical distribution of observations".
> (a) They replace the generator in a GAN by a non-differentiable simulator and solve the minimax problem by minimizing variational upper bounds of the adversarial objectives.
> (b) The main similarity is tuning parameters of a domain-based non-differentiable simulator.
> (c) But they focus on particle physics and in their main experiment, tune a single parameter. Our experiments focus on computer vision and explore a higher dimensional parameter space (11 parameters, 6 for cars, 1 for length to intersection, 4 for weather type).
> (d) Further, while they use policy gradients, they use an adversarial loss to create their reward signal while we use the validation accuracy.
> (e) The most important difference is that we do not seek to mimic the distribution of real-world. In many cases, it is not the distribution that maximizes the reward. In our toy example, we achieve higher accuracy with a learned distribution that is completely different from the ground truth distribution (we add these numbers to the appendix).
>
> Sufficiency of comparison:
> (a) For comparison purposes we believe there are no direct counterparts to our work. The closest related work we have identified is "Learning To Teach" (Fan et al.) since they seek to improve accuracy of a model. Nevertheless, they do not create new data but select which data to train on from existing datasets.
> (b) In order to evaluate our method we present baselines on our experiments. We seek to prove that by learning to simulate we achieve higher accuracy than randomly sampling scenes which is what works such as "Playing for Data: Ground Truth From Computer Games" (Richter et al.) do. We demonstrate this to be the case in all of our experiments.

---

### Official Review · AnonReviewer2 · 2018-11-05
**Great idea, but I don't think the right problems were selected to showcase the method**

**Rating:** 6
**Confidence:** 4

**Review:**

Pros:
* Using RL to choose the simulator parameters is a good idea. It does not sound too novel, but at the same time I am not personally aware of this having been explored in the past (Note that my confidence is 4, so maybe other reviewers might be able to chime in on this point)
* In theory, you don't need domain adaptation or other sim2real techniques if you manage to get the optimal parameters of the simulator with this method.
* Certain attributes of the method were evaluated sufficiently: eg the number of training epochs for each policy iteration, the dataset size generated in each iteration, and whether initialization was random or not in each iteration.
Cons:
* Experiments were underwhelming, and the choice of problems/parameters to tune was not the right one for the problem.
* Parts of the paper could be clearer

QUALITY:
* I believe that although the idea is great, but the quality of the experiments could have been higher. Firstly, better problems could have been selected to showcase the method. I was excited to see experiments with CARLA, but was underwhelmed when I realized that the only parameter of the simulator that the method controlled was the number and the type of cars in the scene, and the task of interest was a car counting task (for which not much detail was provided). This would have been much more interesting and useful to the community if more parameters, including rendering parameters (like lighting, shading, textures, etc) were part of the search space. Similarly, the semantic segmentation task could have used more than one category. But even for the one category, there were no previous methods considered, and the only comparison was between random parameters and the learned ones, where we only see marginal improvement, and what I perceive to be particularly low IoU for the car (although it'd help to know what's the SOTA there for comparison) For both vision applications I could help but wonder why the authors did not try to simply train on the  validation set to give us another datapoint to evaluate the performance of the method: this is data that *is* used for training the outer loop, so it does beg the question of what is the advantage of having hte inner loop.

CLARITY:
* The writing of the paper was clear for the most part, however the experimental section could have been clearer. I was wondering how model/hyperparameter selection was performed? Was there another validation set (other than the one used to train the outer loop)
* The proposed policy is dubbed "its". What does it mean?
* It's not clear what is a "deliberately adversarial" initialization. Could you elaborate?
* The letter R is used to mean "reward" and "rendering". This is confusing. Similarly some symbols are not explicitly explained (eg S) Generally Section 2.3 is particularly unclear and confusing until one gets to the experimental section.
* Section 3 discusses the technique and states that "we can thus generate or oversample unusual situations that would otherwise not be part of the training data" I believe it is important to state that, as the method is presented, this is only true if the "validation" data is varied enough and includes such situations. I believe this would be more applicable if eg rendering parameters were varied and matched the optimal ones.
* Also the method is presented as orthogonal to domain adaptation and other sim-to-real techniques. However, I do not necessarily believe that this paper should be discussed outside the context of such techniques like domain randomization, Cycada, PixelDA etc. Even though these (esp. the latter ones) focus on vision, I do think it sets the right context.
ORIGINALITY:
* As far as I'm aware noone has tried something similar yet. However, I'm not confident on this.
SIGNIFICANCE:
* Although the idea is good, I don't think that the approach to select the simulation parameters presented in the experiments in such a way is significant. I think that eg doing so for rendering parameters would be a lot more powerful and useful (and probably a lot more challenging). Also, I think that a single set of parameters (which seems to be what the goal is in this work) is not what one wants to achieve; rather one wants to find a good range of parameters that can help in the downstream task.

---

> ### Author Response · Authors · 2018-11-25
> **Updated experiments and clarifications on rendering parameters (we do vary rendering parameters in our work)**
>
> We thank the reviewer for appreciating the idea. We hope the following clarifications and experiments allow for a re-evaluation.
>
> Choice of parameters:
> (a) We believe that the paper was unclear about which parameters are learned. Specifically, in Sections 2.3 and 4.2 the reader had been lead to believe that we do not vary the rendering parameters in our work. We do vary lighting parameters in both the car counting and segmentation experiments. We define four weather types - clear noon, clear sunset, wet sunset and rainy sunset. Our policy outputs a categorical distribution over them. The illumination, color hue and light direction are varied as well as reflections from water puddles and weather particles such as rain drops. Sections 2.3 and 4.2 have been modified accordingly.
> (b) We learn not only types of cars and preponderance of cars but also the length of the road ahead, which influence the amount of cars and the structure of the scene.
> (c) We are adding a figure to the appendix to show how weather is learned over time by our method. We observe that our algorithm automatically deduces that giving higher probability to scenes without rain or puddles improves the performance of the main task model.
> (d) We have added text in the paper to highlight the variations described in (a) and (b).
>
> State-of-the-art KITTI segmentation experiment:
> (a) The submission uses ResNet-18 since it is faster for experimentation. We now use a ResNet-50 to achieve a state-of-the-art implementation.
> (b) With ResNet-50, we let our policy learn for 600 iterations and sample random parameters for 600 iterations. Our best policy iteration achieves 0.579 IoU, which is 20% better than the best dataset generated with random parameters (0.480 IoU). Thus, we show a clear improvement over a strong baseline.
> (c) We also introduce another baseline, specifically, random search to optimize over the simulator parameters. Random search achieves 0.407 IoU on the test set. Thus, learning to simulate achieves an increase in performance of 42% over this method. We hypothesize that performance using random search is low due to the nature of the problem which presents sparse and noisy rewards.
> (d) Even though our simulated scenes are limited in their realism, we achieve 57.9% IoU for car segmentation, which is reasonable. As a state-of-the-art reference, an upper-bound of 77.8% IoU is obtained by training the same network on 982 real annotated images, which is much more than the 100 synthetic images used to train our method.
> (e) To make space for these additional experiments we move the dataset size experiments to the appendix.
>
> Use of CARLA:
> (a) Our idea of learning to simulate is independent of the choice of simulator. We choose CARLA to make our contribution concrete. But the resources needed to fully demonstrate on a rich simulator like CARLA are immense. We respectfully submit that such a bar will preclude most groups from publishing on the use of simulators. On the other hand, focusing on a small set of parameters allows more insights into the proposed idea.
> (b) While CARLA is a promising tool, we do extend it in useful ways. It required a significant development effort to turn it into a procedural generator for new traffic scenes. The CARLA plugin is not necessarily built with extensions like this in mind.
> (c) We hope the orthogonal contributions of our paper suggest a useful direction for the CARLA development team too.
>
> Use validation set for training:
> (a) Our use of train-validation-test sets is conventional. It is important to note that we use the validation set akin to hyperparameter selection, rather than using it as labeled training data.
> (b) For deployment, one may follow parameter-tuning with retraining that includes the validation set to achieve the best test performance. However, it’s not common practice for benchmarking new ideas and likewise, we only wish to demonstrate the benefit of learning to simulate.
> (c) There are regimes where the size of validation set is sufficient for evaluation, but not for training. But a small number of real images can instead have a significant impact in terms of bridging the domain gap from simulations, making a fair evaluation tricky.
> (d) Some advantages of learning to simulate persist even if one considers the validation set for training. For example, if a scenario is not sufficiently represented in validation set, it is hard to train a network for it simply by including those images, while our proposed method can oversample it to maximize accuracy.
> (e) As a reference, and instead of training on the validation set, we train on a large real dataset for our main KITTI segmentation experiment.

---

> > ### Author Response · Authors · 2018-11-25
> > **(cont)**
> >
> >
> > Hyperparameters:
> > We use standard hyperparameters for both tasks and use the same ones for all main task networks within an experiment. We use a learning rate of 3e-4 using the Adam optimizer for car counting, as well as standard values for beta_1 (0.9) and beta_2 (0.999). For segmentation, the optimizer used is SGD. We use a learning rate of 6e-4, tuned by generating a balanced dataset containing all weather types, semi-crowded scenes (with cars) and all car types to maximize performance on the KITTI validation dataset. We then use the same learning rate for all synthetic segmentation experiments. To obtain the upper bound trained on 982 annotated real KITTI images, we directly optimize hyperparameters by training on that dataset and using KITTI validation set as a reference.
> >
> > Adversarial initialization:
> > We mean initial parameters that have been chosen to be suboptimal. Specifically, these correspond to using low probability for spawning cars in the scene and higher probability for cars or weather least represented in the test distribution. We modify the paper to be more clear on this point.
> >
> > Notation:
> > We have modified the use of “R” (stylized R for rendering). The proposed policy was named “lts" which stands for "learning to simulate”, but we modified to “LTS" to avoid confusion. Moreover, we have clarified Section 2.3.
> >
> > Oversampling unusual situations:
> > Yes, we need the unusual situation to be present in the validation set, which we assume is representative of test scenarios. While these scenarios are present in the validation set at a low frequency, one does need several samples of rare cases in order to train a network effective for them. That is where oversampling rare scenarios can make a difference.
> >
> > Domain adaptation:
> > We have included extra discussion in Section 3. Note that even if the optimal parameters are learned using our method, there is still need for sim2real domain adaptation. Often the simulator will be limited and not be able to generate images that are completely realistic. Domain adaptation is needed to bridge this gap, thus, leads to orthogonal benefits. The interplay of our method and domain adaptation to achieve stronger results will be interesting future work.

---

### Meta-Review · Area_Chair1 · 2018-12-15
**Methodology could be improved but ideas are very intriguing**

**Confidence:** 4
**Recommendation:** Accept (Poster)

**Metareview:**

This paper discusses the promising idea of using RL for optimizing simulators’ parameters.

The theme of this paper was very well received by the reviewers. Initial concerns about insufficient experimentation were justified, however the amendments done during the rebuttal period ameliorated this issue. The authors argue that due to considered domain and status of existing literature, extensive comparisons are difficult. The AC sympathizes with this argument, however it is still advised that the experiments are conducted in a more conclusive way, for example by disentangling the effects of the different choices made by the proposed model. For example, how would different sampling strategies for optimization perform? Are there more natural black-box optimization methods to use?

The reviewers believe that the methodology followed has a lot of space for improvement. However, the paper presents some fresh and intriguing ideas, which make it overall a relevant work for presentation at ICLR.